# Llama and Alpaca Management in Germany—Results of an Online Survey among Owners on Farm Structure, Health Problems and Self-Reflection

**DOI:** 10.3390/ani11010102

**Published:** 2021-01-06

**Authors:** Saskia Neubert, Alexandra von Altrock, Michael Wendt, Matthias Gerhard Wagener

**Affiliations:** Clinic for Swine and Small Ruminants, Forensic Medicine and Ambulatory Service, University of Veterinary Medicine Hannover, Foundation, 30173 Hannover, Germany; Alexandra.von.Altrock@tiho-hannover.de (A.v.A.); Michael.Wendt@tiho-hannover.de (M.W.); Matthias.Gerhard.Wagener@tiho-hannover.de (M.G.W.)

**Keywords:** questionnaire, llama, alpaca, husbandry, management, health problems, animal welfare

## Abstract

**Simple Summary:**

The keeping of llamas and alpacas is becoming increasingly attractive, resulting in veterinarians being consulted to an increasing extent about the treatment of individual animals or herd care and management. At present, there is little information on the maintenance practices for South American camelids in Germany and on the level of knowledge of animal owners. To gain an overview of the number of animals kept, the farming methods and management practices in alpaca and llama populations, as well as to obtain information on common population problems, a survey was conducted among owners of South American camelids. The findings can help prepare veterinarians for herd visits and serve as a basis for the discussion of current problems in South American camelid husbandry.

**Abstract:**

An online survey of llama and alpaca owners was used to collect data on the population, husbandry, feeding, management measures and health problems. A total of 255 questionnaires were evaluated. In total, 55.1% of the owners had started keeping South American camelids within the last six years. The majority managed small farms with fewer than 15 animals (66.4% of 250 farms). More than half of the participants stated that they kept the camelids as hobby animals (64.3%), although they used them for wool production (55.7%) and/or for breeding (51.8%). Vaccination and deworming were carried out on more than 80% of the farms. The most common diseases occurring on the farms were endo- and ectoparasites. A total of 55.3% of the owners estimated their own knowledge of South American camelids as good, 14.5% as very good. In contrast, more than half of the owners agreed little or not at all with the statement that veterinarians generally have sufficient knowledge about South American camelids. Further research is needed to include veterinarians’ perspectives and thus optimise animal husbandry.

## 1. Introduction

South American camelids include the two domesticated species—llamas and alpacas—as well as the wild forms, guanacos and vicuñas. Molecular genetic studies have shown that llamas (*Lama glama*) and guanacos (*Lama guanicoe*) can be assigned to the genus llama and the species alpaca (*Vicugna pacos*) and vicuña (*Vicugna vicugna*) can be assigned to the genus vicugna [1]. The South American camelids can crossbreed with each other and produce fertile offspring; the crossbred offspring of an alpaca and a llama is called a “huarizo” [2]. Together with the Old World camelids, to which the two species dromedary (*Camelus dromedarius*) and Bactrian camel (*Camelus bactrianus*) belong, the South American camelids are classified in the biological family Camelidae and thus belong to the order of even-toed ungulates (Artiodactyla) and the suborder of Tylopoda [3]. According to archaeological finds and genetic research, the domestication of llamas took place in various places in the Andes of South America. In Argentina as well as in northern Chile, domestication is dated to the years 5000–3800 before present (BP) and in Peru to around 4000 BP. The domestication of alpacas was probably independent and is estimated to have taken place around 6000 BP in Peru [4]. Although at the beginning of domestication South American camelids were mainly kept for meat production and used as transport animals, wool production gained greater importance over time. Since about 1980, the animals have also been kept outside South America [3]. In many European countries, the number of South American camelids being kept is on the rise [5,6,7,8], and in Germany, the keeping of llamas and alpacas seems to be becoming increasingly attractive, leading to veterinarians being consulted more frequently about the treatment of individual animals or herd care and management. This trend seems to be supported if you consider the patient volume of the Clinic for Small Ruminants of the University of Veterinary Medicine Hannover Foundation. About 30 llamas and alpacas were treated in 2010, about 60 animals were hospitalised in 2015 and around 80 South American camelids were treated in 2019 [9]. However, there is currently little information on the keeping of South American camelids in Germany and on the level of knowledge of animal owners.

In Germany, cattle (*Bos taurus*), sheep (*Ovis aries*) and goats (*Capra aegagrus hircus*) have to be clearly identified by numbered ear tags [10,11]. At present, however, llama and alpaca owners do not have to identify their animals or report them to the Animal Disease Fund, but according to § 45 Paragraph 1 of the Ordinance on Protection against the Spread of Animal Diseases in Livestock Traffic (Livestock Traffic Ordinance), there is an obligation to notify the competent authority [12]. However, not all farms are registered [13], and there is no comprehensive information on the types of farming. Although the homepage of the Federal Statistical Office provides concrete information on the number of poultry stocks and the number of cattle (approx. 11.4 million (May 2020)), sheep (approx. 1.6 million (November 2019)) and pigs (*Sus scrofa domesticus*) (approx. 25.5 million (May 2020)) kept in Germany, there is no information on the number of llamas and alpacas [14]. Gauly speaks of an estimated 15,000 llamas and alpacas in Germany [3]. There is no central stock register for South American camelids in Germany, but there are several large associations with registration facilities. In August 2020, 12,458 animals were registered with the German Alpaca Breeders Association (Alpaka Zucht Verband Deutschland e.V.) alone [15].

In order to gain deeper insight into the structures of South American camelid husbandry in Germany, an online questionnaire was conducted among animal owners on husbandry, management, health and self-perception. In the following, the results of this survey are presented and possible conclusions for veterinarians are discussed.

## 2. Materials and Methods

A questionnaire was compiled based on similar studies conducted in Austria and Switzerland [7,16,17]. A total of 69 questions were compiled covering the following five main topics: (1) general information on the farm, (2) husbandry, (3) feeding, (4) management and (5) health problems and medical care (Table 1).

The survey was conducted anonymously and was approved by the Data Protection Officer of the University of Veterinary Medicine Hannover Foundation, Hannover, Germany. It consisted mainly of single and multiple-choice questions and a few open questions. Only the question on federal state location of farms was obligatory, otherwise the survey could be continued even if a question was not answered. The questionnaire was created with the software LimeSurvey (Version 3.13.2 + 180709) from LimeSurvey GmbH, Hamburg, Germany, and participants were invited by various associations (Association of Breeders, Owners and Friends of South American Camelids (Verein der Züchter, Halter und Freunde von Neuweltkameliden e.V.); Alpaka Breeding Association Germany (Alpaka Zucht Verband Deutschland e.V.); Alpaka and Lama Breeding Association Central Germany (Alpaka- und Lama-Zuchtverband Mitteldeutschland e.V.); Alpaca Association e.V.; Working Group European Lama and Alpaka Shows (Arbeitsgemeinschaft Europäische Lama und Alpaka Shows e.V.)) and via social media (Facebook) in February 2020 (opened: 3 February 2020, closed: 23 March 2020). Data analysis was carried out using Microsoft Excel 2016. Statistical tests were conducted using R Statistics version 3.6.1. After excluding participants that had not answered questions up to and including Section 4 (Management), a total of 255 participants were suitable for analyses.

### Statistical Tests

To test whether deworming after a faecal analysis (which we considered a proactive deworming practice) was related to professionality of ownership, a binomial regression was conducted. As only few participants indicated they kept South American camelids as a main occupation, a binary variable concerning whether owners kept their animals for breeding was used as a proxy for professionality.

To test whether the likelihood of gastrointestinal diseases and symptoms was influenced by the professionality of the owners, binomial multiple regressions were conducted. The z-standardised number of animals in the herd and the binary variable of whether owners stated that they owned for breeding (1 = yes, 0 = no) were entered as independent variables. Binary variables of whether or not gastrointestinal parasites, diarrhoea and emaciation had occurred less than once a year (at least) were used as dependent variables. For gastrointestinal parasites an additional independent variable was entered concerning whether owners stated they dewormed their animals following individual or pooled faecal samples, which served as a proxy for proactive deworming practices.

A binomial regression was conducted using the probability that any injuries due to rank fights had been reported by owners (less than once a year at least) as a dependent variable. The z-standardised number of male animals in the herd, the percentage of uncastrated males and the binary variable of whether or not participants owned the animals for breeding were entered as independent variables.

To test whether the likelihood of abortions or stillbirths was influenced by the professionality of the owners, binomial multiple regressions similar to those for gastrointestinal diseases and symptoms were conducted. The z-standardised number of female animals and the z-standardised mean age of female animals, as well as a dichotomous variable concerning whether or not owners had stated that they were using their animals for breeding, were used as independent variables. As dependent variables, the question of whether owners had stated abortions or stillbirths had occurred at least less than once a year was coded as binary.

Testing the assumption that breeders had a higher professionality of ownership than non-breeders, a Welch 2-sample t-test with self-reported own knowledge about South American camelids as a dependent variable (1 = poor knowledge to 4 = outstanding knowledge) was conducted.

## 3. Results

### 3.1. Animal Population

Of the 255 farms evaluated, most were located in North Rhine–Westphalia (49 farms), Lower Saxony (47 farms) and Bavaria (45 farms) (Figure 1).

A total of 55.1% (*n* = 140/254) of the owners had started keeping South American camelids in the last six years (2014–2019) (Figure 2).

A total of 3062 alpacas were recorded on 220 farms and a total of 741 llamas were kept on 75 farms (Table 2 and Table 3). About one third of the male alpacas and more than half of the male llamas were castrated.

A total of 68.6% of all farms (*n* = 175/255) kept only alpacas; another 10.6% (*n* = 27/255) kept only llamas. A total of 16.5% (*n* = 42/255) of all farms kept llamas and alpacas. The remaining farms (3.1%, *n* = 8/255) additionally kept crossbreeding animals (no information: 1.2%, *n* = 3/255).

The 250 farmers who provided data on the number of animals mostly managed small farms with fewer than 15 animals (66.4%, *n* = 166/250). On 3.5% of the farms (*n* = 9/250) fewer than three animals were kept. A total of 30.8% of the respondents (*n* = 77/250) kept between 15 and 50 animals and 2.8% of the farms (*n* = 7/250) had more than 50 animals. Correspondingly, only 8.2% of the respondents (*n* = 21/255) kept South American camelids as a main occupation, 42.0% (*n* = 107/255) had no income and 49.8% (*n* = 127/255) only had additional income from keeping the animals. This is also reflected in the purposes for which animals were kept: participants stated that they kept the South American camelids mostly as a hobby, followed by using them for wool production, breeding and trekking tours. Use of camelids as herd protection for sheep or for meat production was not indicated (Figure 3).

Almost all owners used more than one source of information to inform themselves about keeping llamas and alpacas (98.4%, *n* = 248/252). Books and seminars for further education were mentioned most frequently. More than half of the participants additionally stated that they researched on the Internet (61.9%, *n* = 156/252) or used Internet forums (54.8%, *n* = 138/252) for information, whereas 57.1% of the owners (*n* = 144/252) also referred to information received from veterinarians.

### 3.2. Keeping of South American Camelids

Even though almost half of the animal owners (53.3%, *n* = 136/255) indicated that they kept other animal species on their farms, only 17 of the 255 farms (6.7%) stated that they kept South American camelids in direct contact with other species. Most frequently mentioned was keeping the animals together with poultry. The majority of farms housed the camelids separately from other species (92.2%, *n* = 235/255) (no information: 1.2%, *n* = 3/255).

The distribution of the answers to the question concerning keeping the animals in several groups was proportionately equal (yes: 48.6% (*n* = 124/255); no: 47.5% (*n* = 121/255); no information: 3.9% (*n* = 10/255)). On the farms with several groups, there were mostly fewer than ten animals in one group (58.1%, *n* = 72/124). A total of 34.7% (*n* = 43/124) had a group size of 10–20 animals and 7.3% (*n* = 9/124) kept the South American camelids in groups of more than 20 animals.

Nearly all participants kept their animals in stables or shelters with outdoor access (98.4%, *n* = 251/255). Two participants kept their animals exclusively on pasture without any shelter or in stables (0.4% or one participant each).

The 159 farms with a stable indicated their different floors and bedding materials as multiple answers (Table 4). Bare concrete floors without other bedding materials were used on 13 farms.

The majority of owners provided the South American camelids with several areas set aside for running and used them in a rotation system (78.6%, *n* = 198/252).

### 3.3. Feeding

Almost all herds were fed pasture grass (98.0%, *n* = 248/253) and hay (94.1%, *n* = 238/253) in summer; other basic feed was rarely used.

In winter, hay feeding accounted for the largest percentage of feed (98.8%, *n* = 250/253). Complementary to the feeding of pasture grass (42.7%, *n* = 108/253), the feeding of straw (8.7%, *n* = 22/253) and other feed, especially lucerne and lucerne hay (9.1%, *n* = 23/253), became more important (for further information see the Appendix A). A total of 91.4% (*n* = 233/255) of all farmers fed animals with the basic feed ad libitum. Three owners did not answer this question (1.2%) and 19 owners (7.5%) provided the animals with a limited amount of basic feed. For the limited amount of feed, some owners indicated amounts of 2–4 kg hay per animal per day. 

In addition to the basic feed, mineral feed was offered to the animals on 86.1% (*n* = 216/251) of the farms; 42.6% (*n* = 107/251) of the farms fed concentrated feed to the animals. Eight farms (3.2%) supplied their animals exclusively with basic feed.

Mineral feed was offered ad libitum by some farmers; others limited the quantity. Licks were also provided by around 15 farmers (information from the free-text answers, in some cases unclear). Many also stated that the concentrated feed was only fed to pregnant or lactating females and to weaker animals. Furthermore, some animal owners offered their llamas and alpacas vegetables (9.2%, *n* = 23/251), fruit (5.6%, *n* = 14/251) or other feed (18.7%, *n* = 47/251). Carrots were the most frequently mentioned vegetable, whereas apples were the main fruit.

Out of 235 farmers using concentrated and/or mineral feed in their stock, 151 participants (64.3%) fed their animals vitamin D. The vitamin D dose contained in the feed could be specified by 41 owners and was indicated predominantly in international units (IU). The dose ranged from 1400 IU per kg dry matter to 108,000 IU per kg dry matter.

### 3.4. Management

New additions to the herd were limited to own offspring for the majority of the owners (82.7%, *n* = 211/255), but 14.5% of the farms (*n* = 37/255) regularly introduced new animals (no answer: 2.7% (*n* = 7/255)). According to information from 30 owners, the animals came mainly from different stocks in Germany. More rarely, new llamas and alpacas came from farms in other European countries (*n* = 9/37) or from farms in non-European countries (*n* = 9/37) (multiple choice).

About half of the farms with regular new additions (*n* = 37) initially kept them in quarantine (48.6%, *n* = 18/37). The duration of the quarantine was mostly two weeks (44.4%, *n* = 8/18). Six farms (33.3%, *n* = 6/18) kept new animals separated for four weeks or longer. 

Many farmers possessed proof of origin (pedigree) of their entire South American camelid population (69.0%, *n* = 176/255) or individual animals from their herd (17.3%, *n* = 44/255), although 11.8% (*n* = 30/255) did not have the origin status of their South American camelids or did not answer the question (2.0%, *n* = 5/255). The majority of owners identified their animals by transponders (81.2%, *n* = 199/245) and 1.6% identified animals by ear tags (*n* = 4/245). Four respondents (1.6%) stated that they identified their animals by DNA profile in addition to the transponder (multiple choice). On 18.4% of the farms (*n* = 45/245) the animals were not marked. 

Animals were regularly shorn on all farms. Most animals were shorn once a year (alpacas: 97.7% (*n* = 217/222); llamas: 78.7% (*n* = 59/75)). Only two alpaca owners (0.9%) sheared every other year and one owner (0.5%) stated that the frequency depended on the type of wool. In several llama herds, shearing was done only every other year (10.7%, *n* = 8/75) or in variable time periods (8.0%, *n* = 6/75). Extreme weather conditions and the wool type were mentioned as variables.

On 94.9% of the farms (*n* = 242/255), toenail trimming was carried out, this was mostly done by the owners themselves (91.7%, *n* = 222/242) (multiple choice). On most farms, nails were cut when needed (41.3%, *n* = 100/242) or more than twice a year (38.0%, *n* = 92/242). Only a few owners stated that they cut the nails less frequently (12.8%, *n* = 31/242) (other: 7.9%, *n* = 19/242).

Furthermore, on approximately half of the farms, the fighting-teeth of the adult males were shortened (47.1%, *n* = 120/255). The shortening of the teeth was mainly performed by veterinarians (55.8%, *n* = 67/120) (multiple choice).

According to the participants, 84.3% (*n* = 215/255) of the herds were vaccinated (no: 12.2% (*n* = 31/255); no response: 3.5% (*n* = 9/255)), primarily against clostridial infections (Table 5). Some owners could not specify the vaccination (10.2%, *n* = 22/215).

Vaccination, especially against clostridial infections in general, was mainly carried out once a year (85.6%, *n* = 184/215). Other vaccination intervals were rarely used.

Regarding deworming, the situation was similar to the question about vaccination: 228 of the 255 participants in the survey dewormed their South American camelids (89.4%), 13 owners did not do so (5.1%) and 14 participants did not give an answer (5.5%). Anthelmintics were applied more frequently by injection (66.7%, *n* = 152/228) than orally (52.2%, *n* = 119/228). A few owners (4.4%, *n* = 10/228) stated in the field marked “other” that they alternately dewormed their animals orally and by injection; two respondents mentioned application on the skin as a possible form of application. The frequency of deworming varied considerably from farm to farm (Table 5), but about half of them dewormed depending on the results of an individual or collective animal faecal analysis. Breeders were more likely to deworm following faecal samples than non-breeders, OR = 1.81, z = 2.15, *p* = 0.032.

On 43.0% of the farms (*n* = 98/228), no control of the deworming success was performed. On the remaining farms, a monitoring of the treatment success was carried out by means of parasitological faecal sampling (31.1%, *n* = 71/228), by monitoring the general condition (21.9%, *n* = 50/228) and by monitoring the weight gain (9.6%, *n* = 22/228) (multiple choice; 3.9% without information). The anthelmintic was changed from time to time on 73.2% of the farms (*n* = 167/228).

In 70 stocks (27.5%), additional vitamin and trace element preparations were administered by injection. Table 5 lists the proportion of ingredients used, calculated for the total number of farms. Vitamin B complex and vitamin ADE in particular were added under miscellaneous.

### 3.5. Diseases

The survey asked about the occurrence of various diseases and symptoms in the South American camelids kept. The diseases and symptoms were subdivided according to organ system (digestive system, skin, reproduction, cardiovascular system, musculoskeletal system and miscellaneous). An overview of the frequency of symptoms and diseases can be found in Table 6, Table 7, Table 8, Table 9, Table 10 and Table 11.

The keeping of more animals was related to a higher probability of endoparasites, *OR* = 2.84, *z* = 2.29, *p* = 0.022, of diarrhoea, *OR* = 6.34, *z* = 4.18, *p* < 0.001, and of emaciation, *OR* = 4.55, *z* = 3.83, *p* < 0.001. Breeders reported the occurrence of gastrointestinal parasites with a marginally significantly higher probability than non-breeders, *OR* = 2.17, *z* = 1.96, *p* = 0.050. However, they did not differ significantly for diarrhoea, *OR* = 1.07, *z* = 0.20, *p* = 0.844, and emaciation, *OR* = 1.11, *z* = 0.29, *p* = 0.775. Lastly, deworming after faecal samples was related to a higher probability of gastrointestinal parasites, *OR* = 5.27, *z* = 4.68, *p* < 0.001.

More male animals were related to a higher probability of injuries due to rank fights, *OR* = 5.93, *z* = 4.12, *p* < 0.001, and the percentage of uncastrated males was related to a higher probability of injuries as well, *OR* = 4.61, *z* = 3.26, *p* = 0.001. Breeders had a slightly but insignificantly lower probability of reporting injuries due to rank fights, *OR* = 0.84, *z* = −0.45, *p* = 0.650.

More female animals were related to a higher probability that abortions had occurred, *OR* = 9.91, *z* = 4.43, *p* < 0.001, and that stillbirths had occurred, *OR* = 2.33, *z* = 2.60, *p* = 0.009. The mean age of females did not significantly influence the probability of abortions, *OR* = 1.05, *z* = 0.25, *p* = 0.806, or of stillbirths, *OR* = 1.03, *z* = 0.15, *p* = 0.882. Lastly, owners being breeders did not significantly influence the probability of abortions, *OR* = 0.95, *z* = −0.12, *p* = 0.908, but breeders’ probability of having reported at least some stillbirths was significantly higher than that of non-breeders, *OR* = 4.04, *z* = 2.54, *p* = 0.011.

#### Care by Veterinarians

Almost all South American camelid stocks were cared for by a veterinary practice (87.5%, *n* = 223/255). A few owners consulted different practices (5.9%, *n* = 15/255) or sought no veterinary care (4.7%, *n* = 12/255); 2.0% (*n* = 5/255) did not answer the question. Primarily, mixed practices were consulted (58.8%, *n* = 140/238). Less frequently, the following specialised practices took care of the farms: practice for farm animals (18.5%, *n* = 44/238), specialised practice for South American camelids (9.7%, *n* = 23/238), practice for horses (6.3%, *n* = 15/238), other practices (4.6%, *n* = 11/238), practice for small animals (1.7%, *n* = 4/238). 

The majority of the 238 owners felt well looked after by their current veterinarian and found that this practice had sufficient knowledge in the field of South American camelids (72.3%, *n* = 172/238). The 23 owners who were cared for by specialised South American camelid practices all stated that they were satisfied with the work and knowledge of their veterinarian.

In the last section of the survey, participants were asked to assess their own level of knowledge about South American camelids and to classify the level of knowledge of veterinarians in general. This question included not only their own veterinarian but also other veterinarians. A total of 55.3% of the owners (*n* = 141/255) rated their own knowledge as good, 14.5% (*n* = 37/255) as very good. In contrast, more than half of the owners (*n* = 140/255) agreed little or not at all with the statement that veterinarians generally had sufficient knowledge in the field of South American camelids. Breeders reported higher levels of knowledge (M = 3.04) than non-breeders (M = 2.65), t(246.3) = −4.79, *p* < 0.001.

In addition, the animal owners rated the provision of information material on health problems of South American camelids by veterinarians and veterinary offices as insufficient overall. Veterinarians were negatively evaluated by 56.9% (*n* = 145/255) and veterinary offices by 66.7% (*n* = 170/255) of the owners.

In the last open question, the owners could address topics which were not fully dealt with in the survey. The topic of “care by veterinarians” occupied many owners of South American camelids. Some participants expressed the wish of the owners to have competent veterinarians for questions concerning South American camelid husbandry and -treatment. They also requested an adequate offer of further educational courses and a requirement for proof of attendance of such courses for new owners of llamas and alpacas.

## 4. Discussion

### 4.1. Animal Population

In line with similar research conducted in other European countries, with 3062 animals on 220 farms in Germany, significantly more alpacas are kept than llamas (741 animals on 75 farms). Our research shows an increase in South American camelid maintenance in recent years, as also reported in previous studies [5,7,16,17]. It is remarkable that more than half of the survey participants started keeping South American camelids only within the last six years (2014–2019). The sex distribution differed only slightly between llamas and alpacas. However, it is noticeable that alpacas were castrated less frequently than llamas. A study from England had similar results and suggested that alpacas are used for breeding more often and are generally less dangerous to handle than male llamas [6]. Comparable to population surveys of South American camelids in Switzerland and Austria [7,16,17], small stocks with fewer than 15 animals dominated our survey, with over 90% of owners having no or only additional income through the keeping of South American camelids. As in our study, Kriegl et al. [17] found that the most common uses mentioned were “trekking”, “breeding” and “hobby”. In the present survey in Germany, the keeping of South American camelids for “wool” was indicated similarly frequently. These results support the assumption that llamas and alpacas are mainly kept as hobby animals in Germany. This is additionally confirmed by the fact that none of the 255 survey participants stated that they used llamas or alpacas for meat production. By German law, South American camelids are considered a food-producing animal species [18]. This implies that these animals may only be treated with substances that are listed in Regulation (EU) 37/2010 [19]. In addition, stricter regulations apply to the supply and off-label use of drugs. These restrictions can cause animal welfare problems by complicating the use of drugs by veterinarians. On the other hand, lifting this restriction could also be counterproductive. In the case of non-food-producing species, the procurement and use of drugs is simpler and it is easier for inexperienced animal owners to misuse drugs. Considering the survey results, the classification of South American camelids as food-producing animals should be critically reviewed, but further research is needed.

### 4.2. Keeping of South American Camelids

The legal requirements for keeping South American camelids in Germany are stipulated in the Animal Welfare Act [20]. Llamas and alpacas kept for commercial purposes are classified as farm animals and are therefore also covered by the Farm Animal Husbandry Ordinance [21]. However, both legal documents only provide general guidelines for animal husbandry; defined legal regulations for llamas and alpacas are missing. However, so-called husbandry recommendations can serve as an orientation, such as the expert opinion on minimum requirements for the keeping of mammals from the Federal Ministry of Food and Agriculture Germany (BMEL) [22]. Beyond that, guidelines exist for the keeping of South American camelids in zoos [23,24]. The Veterinary Association for Animal Welfare (Tierärztliche Vereinigung für Tierschutz e.V. (TVT)) has published an instruction sheet for the keeping and use of llamas and alpacas for social purposes as well [25]. Since the minimum requirements published by the Federal Ministry are quite general, the husbandry recommendations by Gauly are also used to assess the current husbandry conditions on the farms of the survey participants [26].

In the minimum requirements of the BMEL, as well as in the recommendations by Gauly, it is recommended to socialise the animals in groups of at least three animals and to keep only one adult male in a group of females [22,26]. Indeed, our analyses show that more male animals and, in particular, more uncastrated male animals were related to more frequent injuries due to rank fights. Although not significantly, breeders reported slightly fewer rank-fight-related injuries overall than non-breeders, indicating that they might be more aware of this problem and more able to keep the animals separated. According to the survey, only a small minority of owners had fewer than three South American camelids and therefore did not implement the recommendations. Nearly half of the owners of llamas and alpacas kept their animals in several groups. It should be noted, however, that especially with regard to the many small farms with fewer than 15 animals, many owners probably kept their South American camelids only in one group instead of several and thus negated the question of keeping them in several groups.

Shared housing with animals of another species is possible if the requirements of the respective species are considered. It should be taken into account, however, that South American camelids are also susceptible to some pathogens, especially parasites of other species (e.g., small ruminants) and that mixed-species grouping can therefore be the cause of a parasite problem [27,28,29]. In the survey, about half of the owners stated that they additionally kept other farm animals in their stock, but mixed-species grouping of the South American camelids only took place on 6.7% of the farms and should therefore scarcely play a role in the parasite burden of the entire population.

In accordance with the guidelines of the BMEL, it is possible to keep llamas and alpacas on pasture all year round, provided that the animals have a shelter or stable. In the stable or shelter, all animals of the herd must be able to be housed. Therefore, 2 m^2^ per adult animal should be available. The outdoor pen should cover an area of at least 300 m^2^ for six adult South American camelids and 25 m^2^ for each additional adult animal. Sand or natural soil is recommended as substrate, and dry and sunny areas should be provided [22]. Gauly and the Veterinary Association for Animal Welfare (TVT) recommend offering a larger area, i.e., an area of 1000 m^2^ for the first two animals and 100 m^2^ for each additional animal should be made available to South American camelids over six months old [25,26]. However, if the predominant nutrient uptake is to take place via the pasture, even larger areas are required. Many different materials are suitable as stable flooring, each with different advantages and disadvantages. For example, sand as a substrate offers a high degree of lying comfort and promotes the natural abrasion of the toenails, but has poor thermal insulation. In this respect, straw and wood shavings offer some advantages, but these materials often cause problems with dust and wool pollution, and wood shavings are also quite expensive. In general, when choosing a substrate, it is important to ensure that the floor is easy to clean and non-slip and that the lying area is as soft and dry as possible [26].

The housing conditions described by the owners seem to be largely in line with the recommendations for South American camelid husbandry. Hardly any owner in the survey reported keeping the animals only indoors or only on pasture. In the stable, straw, rubber mats and wood shavings were mainly used as additional flooring besides concrete. More than half of the farmers had two or even three outdoor areas, and 78.6% of the 252 farmers stated that the animals grazed on their outdoor areas in a rotation system. With regard to the contamination of the areas and thus the animals with parasites, as well as the nutrient supply of the areas, this can be evaluated positively.

### 4.3. Feeding

Since the feed is usually of good quality under Central European conditions, the animals’ maintenance needs can be completely covered by the basic feeds, hay, straw and pasture [22,30]. It should be noted, however, that the energy requirement increases due to additional performance such as growth, pregnancy, lactation or work, and that supplementation with concentrated feed can be useful [30]. Vitamins and minerals are largely absorbed through the basic feed. As determined by the feed used and the differences in the mineral content of the soil depending on the geographical location, it is often advisable to add mineral feed [30,31]. South American camelid owners should regularly determine the nutritional status of their animals using the body condition score (BCS) in order to detect an energetic under- or oversupply [32,33].

All participants in the survey adhered to the recommended basic feed. In summer, pasture grass accounted for the main part of feeding, whereas in winter, alternative feeds were used complimentarily. Hay feeding was carried out all year round by almost all farmers.

In contrast to the survey of South American camelid owners in Austria, where only a relatively small percentage of the farms (39.8%) used mineral feed [16], most owners in this survey indicated supplementing their basic feed with mineral feed (86.1%).

Mineral and/or concentrate feed contained vitamin D in 64.3% of 235 stocks.

Vitamin D is formed in the animals’ skin and in plants in the presence of UV light [30]. The dependence on sunlight is reflected in seasonally varying vitamin D concentrations. In addition, the pigmentation of the animals also plays a role; darker animals often have lower vitamin D concentrations because the UV light essential for synthesis is less easily absorbed. Furthermore, young animals are especially affected by deficiency [34,35,36,37]. It is assumed that South American camelids, due to their origin in the Andes with a significantly higher UV exposure there, have a lower capacity for the self-synthesis of vitamin D. It has been shown that under European conditions, especially in the winter months, vitamin D supplements are sometimes necessary in South American camelids to prevent deficiency symptoms [38,39]. The calculated vitamin D requirement is 30 international units (IU) per kg of body weight per day [40]. Accordingly, a supply of 300 IU/kg dry matter is recommended [30]. In the case of deficiency, an injection at a dose of 1000 IU/kg body weight is recommended in October/November for crias and again in January/February if necessary, and once in January/February for adult females if needed [41]. The correct dosage must be chosen, as fat-soluble vitamins are not excreted when overdosed; these accumulate in the body and can lead to life-threatening damage caused by calcinosis [42,43,44,45].

Many mineral and concentrated feeds for South American camelids already contain vitamin D doses, but the concentration varies quite strongly from feed to feed. A total of 151 survey participants fed concentrates and/or mineral feed with vitamin D supplementation, with the contained dose ranging from 1400 IU/kg dry matter to 108,000 IU/kg dry matter. It can be assumed that it is sometimes difficult to determine the intake of the individual animal in the herd. In addition or alternatively, some herds inject vitamin D preparations, which indicates that there is no uniform approach to vitamin supply among animal owners.

### 4.4. Management

If new animals are purchased, they should initially be kept in quarantine for at least 30 days in order to test the animals for possible diseases and parasites and to prevent these from being carried over into the herd [46]. Required vaccinations should be carried out during this time. This recommended quarantine period was only followed by six animal owners.

Even if a registration of the South American camelids at the Animal Disease Fund and an identification of the animals is not obligatory, the keeping of the animals is subject to the Ordinance on Protection against the Spread of Animal Diseases in Livestock Traffic (Livestock Traffic Ordinance). This ordinance stipulates an obligation to report the stock to the competent authority and to keep a stock register [12]. For the management of the livestock register, as well as for proof of ownership and for pedigree control, a clear identification of the South American camelids is advisable [26], which was considered by the majority of the survey participants.

Care measures should be carried out regularly in the stock. Alpacas should be sheared annually and llamas, depending on the wool type, every 1–2 years (woolly llama) or even less frequently (lightly haired llamas, such as the classic llama) [45]. Toenails should be checked every 2–3 months and cut if necessary. The fighting teeth in males (in rare cases also in females) grow lifelong. In males, it is recommended to shorten these teeth at the age of 2–2.5 years to protect the other animals, as well as the owner [26]. Almost all owners followed the recommendation to shear the alpacas once a year, whereas 10.7% of the farms keeping llamas (*n* = 75) carried out this care measure every two years.

Although South American camelids are susceptible to many infectious diseases that also occur in other farm animals, there are currently no vaccines specifically approved for South American camelids in the European Union (EU) [47]. For the vaccination of llamas and alpacas, off-label vaccines approved for other animal species (often sheep) must therefore be used. The efficacy and tolerability of these have thus not been tested on South American camelids; the dosage must be based on the specifications for the target animal species. Few studies have investigated the use of commercial vaccines in South American camelids. For example, Betancor et al. [48] and Zanolari et al. [49] were able to detect seroconversion in llamas after vaccination with a Clostridia vaccine and in llamas and alpacas after vaccination with a bluetongue virus vaccine. Since South American camelids can suffer from tetanus and enterotoxemia caused by Clostridia, a vaccination against different or single Clostridia species (especially *Clostridium perfringens* type C and D and *Clostridium tetani*) is recommended [45,50]. In this study more than 80% of the owners stated that they vaccinated their animals. This indicates significant progress compared to a vaccination rate of about 40% in 2005 and 2018 in Austria and 2005 in Switzerland [7,16,17]. The majority of the 215 owners answering this question carried out vaccination prophylaxis against Clostridia with a combination vaccine once a year. Due to the fact that Germany is considered to be free of terrestrial rabies since 2008, the result of our survey is quite surprising: 5.1% of the owners had their animals vaccinated against it. The bluetongue virus (BTV) is transmitted by mosquitoes and can also cause serious illness in South American camelids [45]. In a German-wide study from 2008/2009, 14.3% (*n* = 249/1742) of the South American camelids were seropositive for BTV-8 [51]. After Germany experienced a disease-free period from 2012 to 2018, since the end of 2018 sporadic outbreaks have occurred in certain regions. Vaccination against BTV-8 is possible and effective and can therefore be useful in outbreak regions [49]. Vaccination prophylaxis against bluetongue disease was carried out on 7.4% of the farms surveyed. Although no studies on the effectiveness of a vaccination against *Chlamydia* abortion in South American camelids are available so far, an appropriate vaccination was administered in 7.9% of the herds. It is also striking that 10.2% of the owners could not indicate what their animals had been vaccinated against. This demonstrates a lack of knowledge among some animal owners, which veterinarians can help to counteract.

One of the greatest problems in the keeping of South American camelids is the contamination of the stocks with endoparasites, which can be manifested by the occurrence of clinically ill animals and by economic losses for the owners [52,53]. Some endoparasites are host-specific, but the majority are also found in other animal species and transmission between species is possible [52,54]. The parasite burden in a herd depends on many different factors, such as husbandry (stocking density, pasture management, mixed-species grouping) or environmental factors (season, humidity). This should be regularly monitored by means of faecal analysis in order to implement an individual control programme if necessary [45]. Various studies and case reports already prove the occurrence of resistance to certain pharmaceutical agents in South American camelids [55,56,57]. It should also be noted that for llamas and alpacas, there are currently no commercial preparations for deworming available in Germany and off-label use is necessary in any case. In the present study, a similar picture emerged for deworming as for vaccination. The majority of the owners stated that they carried out this measure. This result is positive, as is the fact that more than half of the farmers dewormed depending on the result of an individual or collective faecal sample. In line with our reasoning that deworming after faecal samples can be considered a proactive treatment, breeders dewormed after faecal sample analysis more frequent than non-breeders. This supports the assumption that breeders tend to have more knowledge concerning the management of South American camelids than non-breeders. Accordingly, breeders themselves rated their own knowledge as better. These findings indicate that the provision of knowledge regarding treatment practices might lead to improvements in the treatment of South American camelids. The success of deworming was also checked by some owners, but this check was only carried out in one third by means of a faecal sample analysis. In view of the increasing resistance to anthelmintic drugs, veterinarians should make sure farmers are aware of the potential for drug resistance and establish an appropriate deworming regime.

The vitamin E/selenium supply depends very much on the selenium availability in the soil of different regions. Germany is generally an area of selenium deficiency, and prophylactic application for pregnant females in their last trimester and crias in the first days of life is often appropriate [45]. In the present study, 27.5% of the owners applied vitamin and trace element preparations by injection. The survey does not show to what extent the owners of South American camelids knew and examined the status of their animals regarding vitamin and trace element values in the blood. For this purpose a regular blood test should be carried out by veterinarians.

### 4.5. Diseases

Various retrospective studies deal with the frequently occurring diseases in South American camelids, all the authors of which describe a high percentage of endoparasitosis in llamas and alpacas [58,59,60,61,62]. Other diseases of the digestive tract such as gastric erosions and ulcerations are also common [61,63]. Additionally, diseases of the respiratory tract seem to be of relevance [61]. According to Twomey et al. [62], who investigated 6757 laboratory submissions from South American camelids in England and Wales, “physical emaciation” (weight loss, wasting) was the most commonly reported clinical sign in llamas and alpacas. An often-underestimated problem is the infestation of South American camelids with ectoparasites, especially mange mites. Schlögl et al. [64] examined 326 llamas and alpacas in 13 populations, especially in southern Germany, Italy and Austria, and were able to detect mange mites (*Chorioptes* spp.) in 49% of the cases, but occurring in all 13 populations, by means of adhesive tape method. D’Alterio et al. [65] found a prevalence of 39.8% for *Chorioptes* spp. in southwest England.

According to the information in the questionnaires, diseases and symptoms of disease occurred rarely on the farms. As in the abovementioned studies, diseases caused by gastrointestinal parasites and ectoparasite infestation appeared to be more frequent. This is largely consistent with survey results from Austria and Switzerland, where digestive disorders (endoparasitoses, diarrhoea) and skin issues were also observed in the herds [7,17]. Less than once a year, but still occurring, more than 55 owners also mentioned dental issues, diarrhoea, abortions and emaciation. The animal owners hardly observed or did not observe cardiovascular diseases at all. Overall, the figures seem rather low, and the question arises as to which diseases and symptoms may remain undetected.

It is not surprising that owners with more animals reported observing endoparasites, diarrhea and emaciation more frequently. The assumption that breeders (who may be more experienced owners) have fewer problems with these clinical signs cannot be confirmed. Breeders were even more likely to report having endoparasites in their farm when controlling for the number of animals. However, as stated above, it was also found that breeders dewormed more frequently after faecal samples and were thus possibly able to detect the endoparasites more frequently. Unlike emaciation and diarrhea, endoparasites are probably not detectable by many owners who do not regularly analyse faecal samples, and were therefore reported more frequently by breeders. Abortions were also not observed less by breeders controlled for the number of animals, and stillbirths were even reported slightly more frequently by breeders than by non-breeders. However, it can be assumed that breeders have more births per year overall, thus increasing the probability of abortions and stillbirths. Future research should analyse whether ownership experience and the provisioning of knowledge are able to lower birth-related complications in South American camelid ownership.

The perception of the animal owners regarding their own and the veterinarian’s knowledge was interesting. A good-to-very-good level of knowledge of the owners about llamas and alpacas contrasted with the level of knowledge of the veterinarians in general, which was classified by the owners as predominantly insufficient. Many animal owners were content with their own veterinarian, but obviously had already had negative experiences with previous or other veterinarians. It would be useful to conduct a survey of veterinarians to determine their level of knowledge and compare it to the perception of animal owners. Possibly, the perception is very different and communication would have to be more effective. Further research on this topic is needed.

### 4.6. Limitations

When interpreting the data, it must be borne in mind that the results are based on the subjectivity of the participants. In addition, the data are subject to certain distortions, since, for example, only owners with Internet access could participate in the survey and participation was voluntary. Thus, it remains unclear how representative the current sample is for the whole population of animal owners.

## 5. Conclusions

It can be observed that the keeping of South American camelids is becoming increasingly popular in Germany. Therefore, the treatment of these animals will play an increasingly important role in veterinary practices. It is noticeable that in certain fields, such as vaccination and deworming, there are knowledge deficits among some owners. The owners’ assessment of the level of knowledge of veterinarians in general is an interesting finding, showing that some owners seem to have a poor image of veterinarians and feel inadequately informed by them. Our analyses show that breeders show a higher likelihood of animal treatment according to recommendations than non-breeders and state a higher overall knowledge, implying that knowledge leads to better practice. Thus, future researchers and veterinarians should search for answers as to how knowledge regarding the correct treatment of South American camelids can be more effectively communicated to animal owners.

## Figures and Tables

**Figure 1 animals-11-00102-f001:**
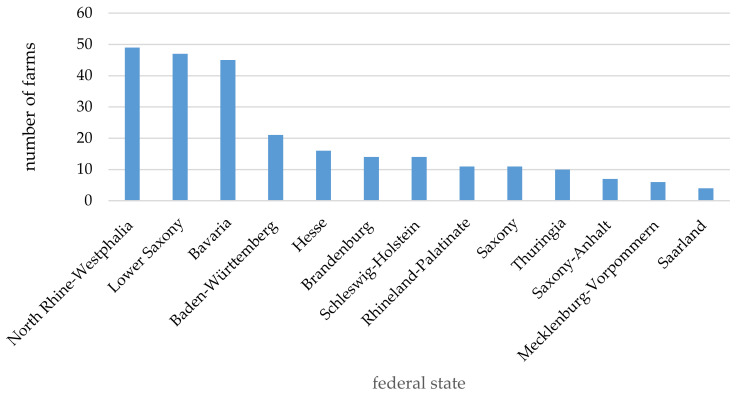
Number of farms by federal state (single choice; *n* = 255).

**Figure 2 animals-11-00102-f002:**
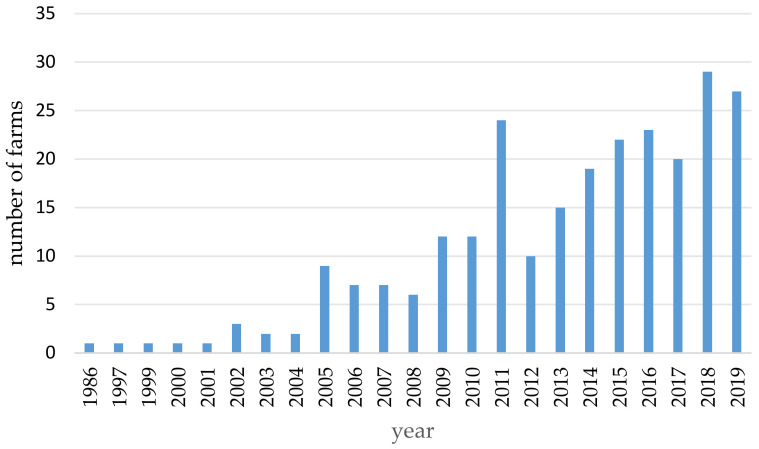
Number of new stocks in the respective year (single choice; *n* = 254).

**Figure 3 animals-11-00102-f003:**
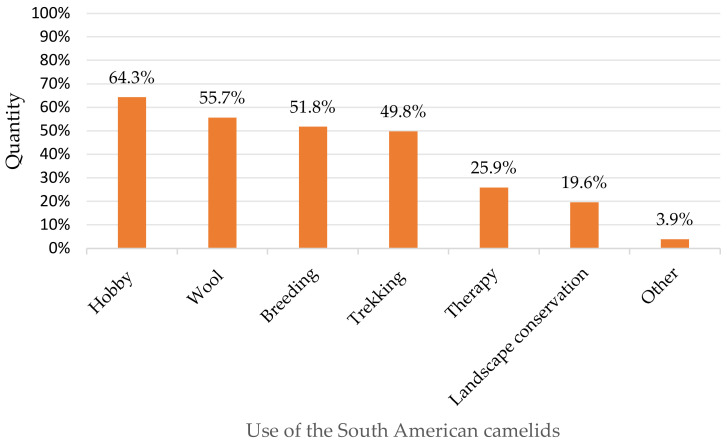
Use of South American camelids (multiple choice; *n* = 255; for explanation “other” see Appendix A).

**Table 1 animals-11-00102-t001:** Summary of the topics covered by the questionnaire.

Section	Topics
General Information	Location of the farm (federal state); species, number, sex and wool type of animals kept; other farm animals kept; start of animal husbandry (year); direction of use; type of income; care of the animals (number of people involved)
Husbandry	Information sources; keeping in groups and group size; stable (size, flooring); pasture/paddock (size, number, rotation)
Feeding	Main feed in summer/winter; feed quantity; additional feed; vitamin D in feed
Management	New animals in the herd (origin, quarantine); proof of origin of the animals; animal identification; shearing (frequency, staff, fixation); cutting the toenails (frequency, staff); shortening the fighting teeth of male animals (staff); vaccination (vaccine, frequency); deworming (application, frequency, monitoring); injection of additional vitamin and trace element formulations
Diseases	Diseases and symptoms of disease (organ system: digestive tract, skin, reproduction, cardiovascular system, musculoskeletal system, miscellaneous); veterinary care (veterinary practice, visit frequency, satisfaction); knowledge about South American camelids (veterinarians, animal owners); availability of information material about South American camelids

**Table 2 animals-11-00102-t002:** Age and sex distribution of alpacas (*n* = 3062) on 220 farms.

Age	Females	Males	Castrated Males	Total
<1	240 (7.8%)	235 (7.7%)	17 (0.6%)	492 (16.1%)
1–3	467 (15.3%)	273 (8.9%)	132 (4.3%)	872 (28.5%)
4–10	830 (27.1%)	305 (10.0%)	268 (8.8%)	1403 (45.8%)
>10	181 (5.9%)	75 (2.4%)	39 (1.3%)	295 (9.6%)
Total	1718 (56.1%)	888 (29.0%)	456 (14.9%)	3062

**Table 3 animals-11-00102-t003:** Age and sex distribution of llamas (*n* = 741) on 75 farms.

Age	Females	Males	Castrated Males	Total
<1	33 (4.5%)	19 (2.6%)	2 (0.3%)	54 (7.3%)
1–3	71 (9.6%)	49 (6.6%)	29 (3.9%)	149 (20.1%)
4–10	192 (25.9%)	65 (8.8%)	117 (15.8%)	374 (50.5%)
>10	75 (10.1%)	34 (4.6%)	55 (7.4%)	164 (22.1%)
Total	371 (50.1%)	167 (22.5%)	203 (27.4%)	741

**Table 4 animals-11-00102-t004:** Floor and bedding in the stables (multiple choice; *n* = 159).

Answer	Farms
Concrete ^1^	104 (65.4%)
Straw	86 (54.1%)
Rubber mats	51 (32.1%)
Wood shavings	30 (18.9%)
Hay	24 (15.1%)
Soil	23 (14.5%)
Other	22 (13.8%)
Wooden floor	12 (7.5%)
Sand	10 (6.3%)

^1^ Bare concrete floors without other bedding materials were used on 13 farms.

**Table 5 animals-11-00102-t005:** Management measures performed in the farms.

Management Measure	Selection	Farms
Vaccinations (*n* = 215, multiple choice)	Clostridial infections in general	169 (78.6%)
	Not known	22 (10.2%)
	Tetanus in particular	17 (7.9%)
	Chlamydial abortion	17 (7.9%)
	Bluetongue disease	16 (7.4%)
	Rabies	11 (5.1%)
	No information	8 (3.7%)
Frequency of deworming (*n* = 228, single choice)	Following pooled faecal samples	78 (34.2%)
	Following individual faecal samples	65 (28.5%)
	Twice a year	42 (18.4%)
	Once a year	26 (11.4%)
	Three times a year	13 (5.7%)
	Less than once a year	3 (1.3%)
	Four times a year	1 (0.4%)
Preparations additionally administered as injection (*n* = 255, multiple choice)	Vitamin E/selenium	57 (22.4%)
	Vitamin D	44 (17.3%)
	Other	12 (4.7%)

**Table 6 animals-11-00102-t006:** Observation of the owners: frequency of diseases or symptoms affecting the digestive system (*n* = 255).

Frequency	Gastrointestinal Parasites	Diarrhoea	Colic	Dental Issues
never	34.9% (*n* = 89)	45.5% (*n* = 116)	63.1% (*n* = 161)	45.5% (*n* = 116)
<1 time a year	29.0% (*n* = 74)	24.3% (*n* = 62)	11.4% (*n* = 29)	26.3% (*n* = 67)
1–2 times a year	19.6% (*n* = 50)	12.2% (*n* = 31)	3.5% (*n* = 9)	7.8% (*n* = 20)
up to 10 times a year	2.4% (*n* = 6)	0.4% (*n* = 1)	0.0% (*n* = 0)	0.4% (*n* = 1)
>10 times a year	0.8% (*n* = 2)	0.0% (*n* = 0)	0.4% (*n* = 1)	0.0% (*n* = 0)
no information	13.3% (*n* = 34)	17.6% (*n* = 45)	21.6% (*n* = 55)	20.0% (*n* = 51)

**Table 7 animals-11-00102-t007:** Observation of the owners: frequency of diseases or symptoms affecting the skin (*n* = 255).

Frequency	Ectoparasites	Abscesses	Mycosis	Injuries due to Rank Fights	Other Skin Issues
never	34.1% (*n* = 87)	52.2% (*n* = 133)	61.2% (*n* = 156)	49.4% (*n* = 126)	45.1% (*n* = 115)
<1 time a year	27.1% (*n* = 69)	20.0% (*n* = 51)	12.2% (*n* = 31)	20.0% (*n* = 51)	18.4% (*n* = 47)
1–2 times a year	23.5% (*n* = 60)	4.3% (*n* = 11)	0.8% (*n* = 2)	9.0% (*n* = 23)	6.7% (*n* = 17)
up to 10 times a year	2.4% (*n* = 6)	0.0% (*n* = 0)	0.0% (*n* = 0)	0.8% (*n* = 2)	0.4% (*n* = 1)
>10 times a year	1.2% (*n* = 3)	0.0% (*n* = 0)	0.0% (*n* = 0)	0.0% (*n* = 0)	0.4% (*n* = 1)
no information	11.8% (*n* = 30)	23.5% (*n* = 60)	25.9% (*n* = 66)	20.8% (*n* = 53)	29.0% (*n* = 74)

**Table 8 animals-11-00102-t008:** Observation of the owners: frequency of diseases or symptoms affecting reproduction (*n* = 255).

Frequency	Abortions	Stillbirths	Difficult Births	Stunted Growth	Malformations in Crias	Lack of Milk in Females	Other Fertility Disorders
never	42.7% (*n* = 109)	47.8% (*n* = 122)	47.5% (*n* = 121)	50.2% (*n* = 128)	56.9% (*n* = 145)	50.2% (*n* = 128)	43.1% (*n* = 110)
<1 time a year	23.1% (*n* = 59)	18.8% (*n* = 48)	13.7% (*n* = 35)	13.7% (*n* = 35)	7.8% (*n* = 20)	12.2% (*n* = 31)	16.1% (*n* = 41)
1–2 times a year	3.5% (*n* = 9)	2.7% (*n* = 7)	5.1% (*n* = 13)	2.0% (*n* = 5)	0.8% (*n* = 2)	3.1% (*n* = 8)	6.3% (*n* = 16)
up to 10 times a year	0.4% (*n* = 1)	0.0% (*n* = 0)	0.4% (*n* = 1)	0.0% (*n* = 0)	0.0% (*n* = 0)	0.0% (*n* = 0)	1.6% (*n* = 4)
no information	30.2% (*n* = 77)	30.6% (*n* = 78)	33.3% (*n* = 85)	34.1% (*n* = 87)	34.5% (*n* = 88)	34.5% (*n* = 88)	32.9% (*n* = 84)

**Table 9 animals-11-00102-t009:** Observation of the owners: frequency of diseases or symptoms affecting the cardiovascular system (*n* = 255).

Frequency	Pneumonia	Heart Murmurs/Heart Diseases	Anaemia
never	71.8% (*n* = 183)	79.6% (*n* = 203)	70.6% (*n* = 180)
<1 time a year	12.9% (*n* = 33)	3.5% (*n* = 9)	9.4% (*n* = 24)
1–2 times a year	0.8% (*n* = 2)	0.4% (*n* = 1)	3.5% (*n* = 9)
up to 10 times a year	0.0% (*n* = 0)	0.0% (*n* = 0)	0.4% (*n* = 1)
no information	14.5% (*n* = 37)	16.5% (*n* = 42)	16.1% (*n* = 41)

**Table 10 animals-11-00102-t010:** Observation of the owners: frequency of diseases or symptoms affecting the musculoskeletal system (*n* = 255).

Frequency	Fractures	Lameness	Exposed Injuries	Limb Deformities
never	76.9% (*n* = 196)	57.6% (*n* = 147)	57.6% (*n* = 147)	67.8% (*n* = 173)
< 1 time a year	8.2% (*n* = 21)	21.6% (*n* = 55)	21.2% (*n* = 54)	12.9% (*n* = 33)
1–2 times a year	0.0% (*n* = 0)	5.9% (*n* = 15)	5.9% (*n* = 15)	0.8% (*n* = 2)
up to 10 times a year	0.0% (*n* = 0)	0.0% (*n* = 0)	0.4% (*n* = 1)	0.8% (*n* = 2)
no information	14.9% (*n* = 38)	14.9% (*n* = 38)	14.9% (*n* = 38)	17.6% (*n* = 45)

**Table 11 animals-11-00102-t011:** Observation of the owners: frequency of other diseases or symptoms (*n* = 255).

Frequency	Emaciation	Recumbency	Central Nervous Disorders	Eye Diseases
never	46.7% (*n* = 119)	69.0% (*n* = 176)	76.9% (*n* = 196)	49.8% (*n* = 127)
< 1 time a year	25.9% (*n* = 66)	9.8% (*n* = 25)	3.5% (*n* = 9)	18.8% (*n* = 48)
1–2 times a year	11.8% (*n* = 30)	1.2% (*n* = 3)	0.0% (*n* = 0)	12.2% (*n* = 31)
up to 10 times a year	1.2% (*n* = 3)	0.0% (*n* = 0)	0.0% (*n* = 0)	0.4% (*n* = 1)
>10 times a year	0.0% (*n* = 0)	0.4% (*n* = 1)	0.0% (*n* = 0)	0.4% (*n* = 1)
no information	14.5% (*n* = 37)	19.6% (*n* = 50)	19.6% (*n* = 50)	18.4% (*n* = 47)

## Data Availability

The data presented in this study are available in Supplementary Material. The additional data other than that are available on request from the corresponding author.

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
