# Peer review of "Llama and Alpaca Management in Germany—Results of an Online Survey among Owners on Farm Structure, Health Problems and Self-Reflection"

_animals, 2021, doi:10.3390/ani11010102_

Round 1

Reviewer 1 Report

I think that the article is very interesting and informative. The only thing I would change is the way it refers to males and females. I don't think that calling them Stallions and Mares is correct and I found it confusing. 

Thanks 

Reviewer 2 Report

The paper will be of interest to a specific group within the scientific community and is of value for publication.

However the survey data was presented in a very basic form with no statistical analysis or knowledge that the participants were a satisfactory representation of the population. More analysis could be done to retrieve more information from the results.

Comments and suggestions to review within the paper;

line 28 should read "parasites" not parasitoses

Line 93 Analysis in excel is not adequate. Some of the data could be analysed using Chi square for significance or a regression analysis - see below suggestions.

line 97 - Is the population and representative sample?

Fig 1 - relevance of farm location? refer to other local Agriculture for reader to understand relevance?

Fig 3 - The percentage does not add up to 100%, probably due to multiple answer Q.

Figure 5 - What is running areas? Do you mean grazing paddocks or rotation grazing?

Line 200 - Should be "shorn" not "sheared"

Line 229-233 Monitoring of worms and checking success of treatment is a positive result, but this is not discussed in the paper.

Table 5 - Correlation between observation of disease with those participants using anthelmintics would be interesting result. This would show if owners are proactive. They appear to be as 31% check the success of treatment.

Table 6 - Correlate the injuries from fighting with owners keeping stallions vs geldings?

Table 7 - Abortions at 23% is very high - Is this a factor of age of animals? Correlate experience and knowledge of owners wit abortions?

Table 8 - Correlate cases of pneumonia with location.

Table 9 - lameness? need further information on relationships to lameness

Line 263 - 72.3% of responses felt vets looked after animals with sufficient knowledge. However in Line 270 50% thought vets didn't have sufficient knowledge - ambiguous.

Figure 13 - difficult to interpret as it doesn't add up to 100%.

More in depth discussion required.

Reviewer 3 Report

Animals review – llama and alpaca management

Dear authors,

Thank you for the opportunity to review this paper. Overall I think it read well and you seem to have collated a lot of data. I think there are sections in the discussion where you could expand on the points you make and please be mindful of your recommendations to the veterinary profession. I don’t think they are appropriate based on the data you collected. The results are very ‘figure heavy’ and I have recommended removing a lot of these where possible and replacing with text. Finally, I think the conclusion could more clearly reflect your work and the importance of it, and would therefore recommend reworking this. I hope my comments are helpful.

L31 – 33: quite a bold statement! Maybe it could be toned down to relate to owner perceptions of veterinary knowledge, or did you survey veterinarians and ask them what they thought they knew about the topic?

L53: do you have a reference to support the statement that keeping of llamas and alpacas is increasing in Germany? Likewise, for the statement in relation to veterinarians being consulted more frequently?

L58: latin names needed (and throughout where new species is introduced)

Table 1: I would suggest saying ‘group size’ rather than ‘keeping in groups’. Why has ‘additional vitamin and trace element formulations’ been put under management rather than feeding?

Figure 1: title is not needed on this graph. It would look better if numbers were removed (as the bars indicate these). Remove names of states if no survey respondents were from there. I would also remove the red colouration from the first three bars. [graph formatting comment re titles and numbers above bars is the same throughout]

Figure 3: again remove purposes which do not have any responses. Remove % from the numbers on y axis – this is covered in the y axis title. Other comments as above

L111: would suggest just saying ’… farms kept only alpacas’ rather than ‘stated to keep’

L113: what do you mean by ‘no information’? is that number of respondents that didn’t answer the Q? I would suggest being consistent with how you deal with this throughout the MS. It may be easiest to say at the start of each section that XX respondents answered this Q. and then express %ages as the % of the people that answered that Q in that manner

Figure 4. You have a lot of figures in this paper so I would suggest including this information in the text and getting rid of this figure as it is quite simple.

Figure 5. Again remove and incorporate this information in the text

Figure 6. As there isn’t much fed beyond pasture/hay I would also consider incorporating this information into the text and directing the reader to the supplementary files for further information

L176: that is a very long way of saying that 151 farmers fed food containing vitamin D!

L177: what is the difference between negating the question and not answering the question?

Figure 7: I would remove and summarise in the text

Figure 8: as above

Could figures 9 – 11 be combined and presented in a table format instead? I think it would be easier to see all of that information in one place rather than spread out

L199: I think this sentence could be simplified to say ‘animals were regularly sheered on all farms’

L217: which vaccination?

Figure 12: I think these can be removed and the detailed information included in supplementary material. I am not clear (L268) how you are expecting people to pass judgement on vets ‘in general’? I think it is sufficient for them to be talking about their own practice rather than places they presumably haven’t visited.

Figure 13: I think this can also be incorporated in the text

L309-310: why should it be reviewed? Just because not many people (from those that you surveyed) are keeping them for meat? Or because owners aren’t following guidance in relation to substance use?

L328: nearly half of your survey population is a reasonable proportion – I would probably consider removing ‘just’ from this sentence

L336: suggest changing the word ‘socialisation’ for ‘mixed-species grouping’ or something along those lines (be mindful of this throughout – I interpret ‘socialisation’ as an interaction that goes beyond merely being in the same housing area as another species or individual)

L358: remove ‘his’ – either just remove it or replace it with a gender neutral term

L361: I think this sentence could be expanded to briefly say why this type of management is positive

Discussion: you don’t need to include specific percentages in your discussion as these are all covered in the results. You also don’t need to refer back to figures that were presented in your results in the discussion

L452: why are you assuming that they were inadequately advised by vets? I think that is quite a dangerous assumption, it may be that they don’t have a record, or they have forgotten. I would temper down statements like this as you don’t really have evidence to support them

L459: do you mean here with other animals of the same species or with other species?

L465: that they would carry it out or that they do carry it out?

L468: I don’t think sensitise is the correct word here – maybe make sure farmers are aware of the potential for drug resistance?

L477: are you recommending a regular blood test be something that farmers do? If so then I would make it clear this is a recommendation

L485: symptom of what? Or are you just saying physical emaciation was the most commonly reported problem? If so then just rewrite the sentence without the use of the word symptom

L501 (and following paragraph): again be mindful about recommendations that you are making to the veterinary profession on the whole based on no real evidence. Maybe instead of what you have currently put it would be more appropriate to conduct a survey of veterinary professionals to identify their knowledge base, and the compare that to customer perceptions. If then there are differences between the two there is a need for more effective communication. I would suggest in some instances owners maybe just do not like what the vet is saying

L515: suggest changing ‘more and more’ to ‘increasingly’

L519: I really strongly advocate removing this sentence, many of your owners were happy with their vets knowledge, and vets already undertake a lot of training anyway! You should be very careful with this angle – you don’t know what information owners were given and what they did with that information

L525: you are basing your recommendation on a survey of ~250 people, what is this like in terms of proportion of the potential population? What are the ramifications of such a recommendation? What are the issues with retaining drug regulations as they are?

Conclusion: I think this needs reworking to be a better reflection of the work you undertook, rather than recommendations based on a lack of evidence
